REGISTERED REPORT PROTOCOL

# *Morus alba*: Host reaction for *Meloidogyne javanica*, biological nematicides assessment and study of these relationships with yield and quality of leaves, cocoon and health of the silkworm

**Santino Aleandro da Silva**[1,2☯], **Renata da Rosa**[1☯], **Rachel Colauto Milanezi-Aguiar**[1☯], **Cristianne Cordeiro Nascimento**[3☯], **Andressa Cristina Zamboni Machado**[2☯]*

**1** Centro de Ciências Biológicas, Universidade Estadual de Londrina -UEL, Londrina, Paraná, Brazil, **2** Department of Plant Pathology, Instituto de Desenvolvimento Rural do Paraná –IAPAR/EMATER, Londrina, Paraná, Brasil, **3** Centro de Educação, Comunicação e Artes, Universidade Estadual de Londrina– UEL, Londrina, Paraná, Brazil

☯ These authors contributed equally to this work.
* andressa_machado@idr.pr.gov.br

## Abstract

Root-knot nematodes cause damage to several crops and the importance of each species can vary according with the crop and the agricultural region. In Brazil, *Meloidogyne javanica* is one of the most important nematode species parasitizing mulberry. To define management strategies, it is important to know if the crop species is damaged by the parasitism of the nematode and the best choices for control, as the use of nematicides. Biological nematicides have been extensively used in Brazil, but no information regarding its efficiency to control *M. javanica* in mulberry is available. Besides, it is not known if biological nematicides could improve the quality of leaves or if they alter the nutrient composition of leaves, which could interfere in the development of the silkworms that are feed with these leaves or in the quality of the silk produced. With the aim to address these questions, we propose a study that will start in the phenotyping of the main Brazilian mulberry cultivars to *Meloidogyne* species, passing through the test of efficiency of biological nematicides in the control of *M. javanica* in mulberry cultivar Miura, evaluation of the amount and quality of leaves produced and, using these leaves to feed silkworms, in the analyzes of the impact of these diet in the health of silkworms, and in the production and quality of the silk.

## Introduction

Brazil is the fifth world main producer of silk, in which Paraná State is the most important Brazilian producer, where sericulture occupies about 3,000 hectares, distributed in 1,700 farmers [1, 2], and set up as an activity highly related to familiar agriculture. Sericulture involves the crop of mulberry leaves (*Morus alba*), the silkworm (*Bombix mori*) cultivation and the spinning. The quality of silk, as well as the yield of cocoons, is directly related to nutritional factors

reproduction in any medium, provided the original author and source are credited.

**Data Availability Statement:** All relevant data from this study will be made available upon study completion.

**Funding:** Initials: RdR - grant number: TC 055/2018 - Funder: Governo do Estado do Paraná (Unidade Gestora do Fundo Paraná). Initials: RdR - Funder: Fiação de Seda Bratac S.A. Initials: RdR - grant number: 302770/2017-8- Funder: Conselho Nacional de Desenvolvimento Científico e Tecnológico (CNPq) Initials: ACZM - grant number: 308814/2018-5 - Funder: Conselho Nacional de Desenvolvimento Científico e Tecnológico (CNPq).

**Competing interests:** No authors have competing interests.

and to the adequate management of the mulberry leaves offered to the silkworms, which must be free of any agrochemical residues [3–6]. By this way, the volume and the quality of the mulberry leaves are of extreme importance to the success in obtaining high quality cocoons.

An important disease which affects the crop is that caused by root-knot nematodes from the genus *Meloidogyne*. Several *Meloidogyne* species have been reported parasitizing mulberry worldwide. In China, *M. enterolobii* [7], in Japan, *M. mali* [8], in India, *M. incognita* [9], in Brazil, *M. incognita* and *M. javanica* [10], and, also in Brazil, *M. enterolobii* was described parasitizing *Morus nigra* (blueberry) [11]. These reports highlight the importance and distribution of root-knot nematodes in mulberry growing regions worldwide.

Plants parasitized by *Meloidogyne* spp. can have low photosynthetic rates and in long term experiments, which include the complete vegetative cycle of the crop, it is possible to infer about the losses related to the nematode parasitism. Besides, the parasitism by *Meloidogyne* spp. can lead plants to hydric stress [12] and to severe alterations in the root system of mulberry [13]. Moreover, the higher *M. incognita* incidence the higher the presence of other root and leaf diseases and plant depletion [9]. In a study conducted with mulberry plants cropped in microplots infested and non-infested with *M. incognita*, it was verified reduction in the number of leaves, weight and budding of infested plants in relation to non-infested [14].

Management of nematodes in mulberry represents a challenge either by it is a perennial crop or because of the possible risks of leaf contamination with chemical residues, since leaves will be offered in the silkworm diet and contaminations can interfere in the worm development. Besides, in Brazil there are no chemical nematicides registered by Brazilian Ministry of Agriculture, Livestock and Food Supply for use in mulberry fields. So, the use of biological nematicides is an important option, since they are registered for the biological target and not for crops in Brazil. By this way, two biological control agents that could be used in the management of *Meloidogyne* spp. in mulberry are *Pochonia chlamydosporia*, a parasite of eggs and juveniles, and *Bacillus firmus*, a bacteria which colonizes the rhizosphere forming a biofilm that causes disorientation in the nematode, avoiding the penetration of nematodes into the roots.

Considering available information until now, we verified that *M. incognita* is the more studied nematode species in mulberry worldwide, but, in Brazil, it was pointed out that *M. javanica* is the main species found in mulberry fields in Paraná State [10]. Therefore, our study aiming to evaluate the effect of biological nematicides in mulberry plants inoculated with *M. javanica* in the development of mulberry plants and, later, to evaluate the effects in the growth of silkworms feed by these leaves.

In the first phase of this project, the three main cropped mulberry cultivars in Brazil, Miura, Tailandesa, and Taichi, will be phenotype in relation to their reaction to *M. incognita* and *M. javanica*, the main nematode species occurring in Brazil in mulberry [10]. In the sequence, we will evaluate two biological nematicides, *P. chlamydosporia* (Rizotec®) and *B. firmus* (Votivo Prime®) in the control of *M. javanica* in mulberry cultivar Miura. Following, with the cultivar Miura, indicated as susceptible to *M. javanica* [10], we will test the effect of these nematicides in inoculated and non-inoculated plants with *M. javanica* in the yield and protein and mineral content of leaves. Leaves from each treatment obtained in this experiment will be offered to silkworms during the corresponding period to the cultivation time normally carried out by sericulture farmers, under controlled conditions, and at the end of the cycle, production of silk will be measured, as well as the gene expression levels of the main genes involved in the silk production. In addition, histological analyzes of the digestive tract and sericigenic glands will be done to verify the cell integrity and worm healthy after treatments.

By this way, in the final of this project, we will have a broad overview of the real impact of *M. javanica* in the silk production, from the productive base, represented by the mulberry tree, through the silkworms and the silk extracted from the cocoons.

## Material and methods

### Mulberry seedlings obtention

Seedlings will be produced from cuttings taken from mature plants, with buds developed, collected from the germplasm collection maintained by Instituto de Desenvolvimento Rural do Paraná –IDR–IAPAR/ EMATER, located at the municipality of Londrina, Paraná State, Brazil. 10 cm-length cuttings from the cultivars Miura, Taichi, and Tailandesa with one bud each will be cut and maintained on water during 24 hours to hydration. After, cuttings will be transferred to 945 ml-capacity Styrofoam pots fulfilled with a mixture of soil and sand in the proportion of 75% of sand, 5% of silt, and 20% of clay, previously sterilized under forced air circulation drying chamber at 150 °C for 5 hours. Each seedling will be fertilized at the time of planting with 5 g of Osmocote Plus (15 N; 9 P; 12 K; 1 Mg; 2.3 S; 0.05 Cu; 0.45 Fe; 0.06 Mn; 0.02 Mo) (ICL Specialty Fertilizers).

After planting the cuttings, pots will be maintained under humidity chamber for 30 days, with programed nebulization. After budding and rooting, seedlings will be transferred to a greenhouse to acclimatization for 15 days.

### Host reaction of mulberry (*Morus alba*) cultivars to *Meloidogyne* spp.

Seedlings of the three mulberry cultivars, after acclimatization, will be tested against *M. incognita* and *M. javanica* to confirm previous report about their host reaction to these root-knot nematodes. Experiment will be conducted in completed randomized design, with six replicates per treatment, in which each pot containing one mulberry seedling will be considered the plot, in factorial arrangement 3x2 (three mulberry cultivars x two *Meloidogyne* species).

Inoculations will be done using pure populations of each nematode species, extracted from tomato roots through the methodology of trituration-sieving described in literature [15]. Nematodes extracted will be quantified under light microscope with the aim of a Peters slide and suspensions will be adjusted to contain 500 eggs per ml. Inoculation will be performed by pipetting 2 ml of each suspension per plant, distributed in two 4 cm holes made near the plant stem, individually, resulting in an initial population density of 1,000 eggs per plant.

Evaluation will be done after 120 days from inoculation (DAI), when roots will be washed free of soil, weighted, and submitted to nematode extraction [15]. Nematodes extracted from each plant will be quantified to obtain the final population (FP) that will be divided by the initial population (IP) to calculate the reproduction factor (RF = FP/IP) of the nematodes [16]. Plants with RF values higher than 1.0 will be classified as susceptible to the nematode and those with RF values lower than 1.0, as resistant. The number of nematodes per gram of roots (nema.g) will be obtained dividing the total number of nematodes present in each plant by the fresh root weight. This variable will be submitted to the homogeneity of variances test of Bartlett and to the normality of data test of Shapiro-Wilk. To attend the assumptions of the model, the transformation of data will be done according to Box-Cox procedure, followed by the two-way variance analysis, when the unfolding of interactions will be performed and means will be grouped by the Scott-Knot test.

### Evaluation of the impacts of the biological nematicides in the yield and quality of mulberry leaves cultivar Miura

Acclimatized mulberry seedlings will be transplanted to 5 liters-capacity plastic pots containing the same mixture of soil and sand. Experiment will be conducted in completed randomized design, with fifth replicates per treatment, in which each pot containing one mulberry seedling will be considered the plot. Treatments will be as follows: 1) absolute check (without nematode and without nematicide); 2) inoculated check (with nematode and without nematicide); 3)

with nematode and *Pochonia clamydosporia* (Rizotec®); and 4) with nematode and *B. firmus* (Votivo Prime®).

Fifth days after the transplant, 5,000 eggs of *M. javanica* will be inoculated per plant in treatments 2, 3, and 4. Ninety DAI, a second fertilization with 5 g of Osmocote Plus per plant will be performed and, after five days, mulberry plants will be pruned and biological nematicides will be pulverized in soil, in bands around each side of plants. After more 75 days, leaves will be collected to feed silkworms. In addition, leaves will be collected to tissue analysis aiming to verify the nutrient and crude protein contents. After cutting of leaves, plants will be submitted to a destructive analysis to verify the nematode multiplication, according with the described earlier.

Data regarding the multiplication of the nematode (RF and nema.g) will be submitted to the univariate analyzes, as follows: data will be submitted to the homogeneity of variances test of Bartlett and to the normality of residuals test of Shapiro-Wilk. To attend the assumptions of the model, the transformation of data will be done according to Box-Cox procedure, followed by the one-way variance analysis, and means will be compared by the Tukey Honestly Significant Difference test.

Data regarding the nutrient content of leaves (nitrogen, phosphorus, potassium, calcium, magnesium, sulfur, boron, cuprum, iron, manganese, zinc, molybdenum, nickel, aluminum, silicon, cobalt, sodium, selenium, vanadium) will be analyzed by multivariate analyzes, with principal component analysis based in the correlation and means grouping by dendrogram based in cophenetic matrix, using the Euclidian distance and the average method to clade obtention. Bromatological analysis of the leaves will be also performed, using the variables crude protein and dry matter, by one-way variance analysis and means will be compared by the Tukey Honestly Significant Difference test as described above.

## Silkworm (*Bombix mori*) creation and feeding

Third-stage larvae of *Bombix mori* (240 larvae) from the commercial hybrid ♂Japanese x ♀ Chinese (provided by Fiação de Seda Bratac S. A.), with the same age (9 days), weight and parental origin will be maintained in creation chambers with temperature, humidity and photoperiod controlled (27 °C; ±80% RH; 14 hours light) for this study. Larvae will be aleatory divided in four experimental groups (80 per group) of feeding: 1) feeding with fresh leaves collected in plants without nematode and without nematicide; 2) feeding with fresh leaves collected in plants with nematode and without nematicide; 3) feeding with fresh leaves collected in plants with nematode and Rizotec®; 4) feeding with fresh leaves collected in plants with nematode and Votivo Prime®. Each group will receive the same amount of leaves, according with recommendations from literature [5, 17].

Larvae will be maintained in plastic trays (50 cm x 40 cm x 10 cm) and fed four times at a day (7:30 am, 11:30 am, 3:30 pm, and 7:30 pm) until they complete their larval cycle and reach the cocoon stage. Trays will be covered with tulle tissue to allow aeration. Trays will be cleaned and feces will be removed daily in the early morning, before the first feeding. Values about the food consumption of each group will be calculated daily, through the weight of the leaves offered, leaves not consumed and the weight of the feces. In addition, the average body weight per group and the mortality rate at the beginning and end of each instar will be also measured. Data regarding the variables will be submitted to the univariate analyzes described earlier. All analysis will be performed with R software [18], using the packages MASS [19], ExpDes [20], and vegan [21].

## Histological analysis of intestinal and glandular tissues of *Bombix mori*

**Light microscopy.** In the fifth day of the last larval stage, five larvae from each group will be randomly selected, anesthetized and dissected in saline solution for insects (1.80 g NaCl;

1.88 g KCl; 0.16 g CaCl2; 0.004 g NaHCO3; 100 ml distilled water) aiming to collect the midgut and sericigenic glands. These tissues will be fixed in Karnovysky fixative for 24 hours and after, maintained in ethylic alcohol 70% until the time of inclusion and processing. The inclusion process will be made in historesin Leica®, followed by these procedures: dehydration in ethylic alcohol 80% for 20 minutes; dehydration in ethylic alcohol 95% for 20 minutes; soaking in a mixture of historesin and ethylic alcohol 95% for 4 hours; soaking in pure historesin for 12 hours; and, finally, soaking and inclusion in pure resin added with catalyzer using appropriate models. After polymerization in incubator at 37 $^{o}$C for 24 hours, blocs will be adhered with Araldite® on wooden supports.

Histological cuts will be performed in a thickness of 4 μm with a microtome Leica®, placed on a hot plate for better adherence to the slides and submitted to staining with hematoxilin and eosin (HE), according with the following described: distilled water for 10 minutes; staining with Harris´ hematoxilin for 20 minutes; tap water for 10 minutes; staining with eosin for 25 minutes; wash with distilled water; ethylic alcohol 90% for 10 minutes; ethylic alcohol 100% for 10 minutes; xylol I for 10 minutes; xylol II for 10 minutes; and xylol III for 10 minutes. After, slides will be assembled in Entellan® medium and analyzed under a microscope Zeiss Axiophot equipped with a camera and Motic software.

**Transmission electron microscopy.** Tissues will be fixed in glutaraldehyde 2.5% in phosphate buffer solution 0.1M (pH 7.2) for 8 hours, post-fixed in osmium tetroxide 1% in the same buffer for 2 hours, contrasted in blocks with uranyl acetate 0.5% for 2 hours, dehydrated and included in historesin. Ultrafine cuts (50 nm) will be performed, contrasted in alcoholic solution saturated with uranyl acetate for 20 minutes and lead citrate for 20 minutes. Slides will be analyzed in an electronic microscope FEI TECNAI 12, in which the tissue integrity and morphological patterns of the epithelial tissues will be observed.

## Real-time PCR analysis (qRT-PCR)

The mRNA expression patterns of genes related to silk production (sericin and fibroin) in silkworms fed with leaves treated with the different treatments will be evaluated by real-time quantitative PCR (qRT-PCR). Three samples will be used in triplicates for each experimental group of larvae. The tissue samples will be previously collected and immediately freezes in liquid nitrogen and then stored at -80˚C. Total RNA from each sample of the medium silk gland (~ 0.5g of tissue) will be isolated using PureLink$^{TM}$ RNA Mini Kit (Invitrogen) according to the manufacturer's specifications. The RNA samples will then be stored at -80˚C. The quality of the RNA will be evaluate using BioDropTM (Isogen Life Science) and Agilent 2100 Bioanalyzer (Agilent Technologies).

The primer pairs (Table 1) that will be used in qRT-PCR are *Sericin-1*, H-*Fibroin*, L-*Fibroin*, were designed using Mega-X and Gedit softwares and will be synthesized by Thermofisher Scientific (Invitrogen, Londrina, Paraná, Brazil). The internal reference genes, ribosomal protein L3 (RpL3) and alpha-tubulin (α-tubulin) that will be used were described in literature [22].

The primers were designed using genes sequences previously deposited in the NCBI for *B. mori* and *B. mandarina*. The sequences were searched using the follows terms: "silk sericin", "silk heavy fibroin", "silk light fibroin". After that, we preferentially selected sequences related to the *B. mori*, such as Sericin 1 (M26105.1, JX681123.1, Z48802.1, AB112020.1 and NM_001044041.2), H-Fibroin (DQ459410.1, AB017362.1, S74439.1, XM_028178929.1 and AF226688.1), L-Fibroin (NM_001044023.1, X17291.1, AY078388.1 and HQ116534.1).

The cDNA synthesis will be performed in a T100TM Thermal Cycler (BIO-RAD), using 12.4 μl of total RNA (1000 ng/ml), 1 μl of oligo dT (20 pmol/ml) and 2 μl of dNTPs (2.5 mM) for each sample, which will be maintained for 10 min at 65˚C. Samples will be cooled quickly

**Table 1. Sequences of primers used in qRT-PCR.**

| Gene Name | Primer Sequences (5'-3') | Product Size (bp) |
|---|---|---|
| Sericin 1 | CTACCGTACAGTCATCCAC | 192 |
|  | GTCTGTTCACGCCTGAATG |  |
| H-Fibroin | ACAAGGTGCAGGAAGTGC | 144 |
|  | GCAATTCACACAAGGCAGT |  |
| L-Fibroin | CCGGAGGTGGAAGAATCTAT | 155 |
|  | GGTTATGTAGGCAGCGATGT |  |
| α-tubulin | ACATGGCTTGCTGTATGCT | 146 |
|  | GGGTGGCTGGTAGTTGATA |  |
| RpL3 | CGGTGTTGTTGGATACATTGAG | 161 |
|  | GCTCATCCTGCCATTTCTTACT |  |

by placing microtubes on ice and, then, 4 μl of MLV 5x First Strand Buffer (M-MLV–Invitrogen, Life Technologies) [250mM Tris-HCL (pH 8.3), 375 mM KCl, 15 mM Magnesium Chloride] will be added to each sample for 2 minutes. So, 0.1 μl of RNAout (Invitrogen, Life Technologies) and 0.5 μl of reverse transcriptase enzyme (M-MLV–Invitrogen, Life Technologies) will be added to each sample and the reactions will be maintained at 37˚C for 50 minutes, 70˚C for 15 minutes and, then, maintained at 10˚C. qRT-PCR will be conducted in an Applied Biosystems® Real-Time PCR Systems, using 6.0 μl of Platinum SYBR Green qPCR Supermix-UDG (Invitrogen), 0.5 μl of each pair of oligonucleotides, and 5 μl of cDNA of each sample. The qRT-PCR reactions will be performed under the following conditions: 95˚C for 5 minutes and 45 cycles [95˚C/ 15 s, 60˚C/ 20 s, 72˚C/ 20 s]. The melting curve analysis will be used to analyze the specificity of the qRT-PCR product. The qRT-PCR will be conducted with two technical and three biological replicates.

Data will be normalized to the expression values of RpL3 and α-tubulin constitutive genes. Analyses of the relative expression and normalization for constitutive genes will be tested with pairwise fixed reallocation randomization test [23, 24], with a level of significance fixed at $p < 0.05$.

# Author Contributions

**Conceptualization:** Santino Aleandro da Silva, Renata da Rosa, Cristianne Cordeiro Nascimento, Andressa Cristina Zamboni Machado.

**Data curation:** Santino Aleandro da Silva, Rachel Colauto Milanezi-Aguiar.

**Formal analysis:** Santino Aleandro da Silva, Renata da Rosa, Rachel Colauto Milanezi-Aguiar.

**Funding acquisition:** Renata da Rosa, Cristianne Cordeiro Nascimento.

**Investigation:** Santino Aleandro da Silva, Renata da Rosa, Rachel Colauto Milanezi-Aguiar, Andressa Cristina Zamboni Machado.

**Methodology:** Santino Aleandro da Silva, Renata da Rosa, Rachel Colauto Milanezi-Aguiar, Andressa Cristina Zamboni Machado.

**Project administration:** Renata da Rosa, Andressa Cristina Zamboni Machado.

**Resources:** Renata da Rosa, Cristianne Cordeiro Nascimento, Andressa Cristina Zamboni Machado.

**Software:** Santino Aleandro da Silva.

**Supervision:** Renata da Rosa, Cristianne Cordeiro Nascimento, Andressa Cristina Zamboni Machado.

**Validation:** Santino Aleandro da Silva.

**Writing – original draft:** Santino Aleandro da Silva, Renata da Rosa, Rachel Colauto Milanezi-Aguiar, Cristianne Cordeiro Nascimento, Andressa Cristina Zamboni Machado.

**Writing – review & editing:** Santino Aleandro da Silva, Renata da Rosa, Rachel Colauto Milanezi-Aguiar, Cristianne Cordeiro Nascimento, Andressa Cristina Zamboni Machado.

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
