## [Decision Letter · Decision Letter 0]

1 Apr 2021

PONE-D-20-23665

Morus alba: phenotyping for Meloidogyne javanica, biological nematicides assessment and study of these relationships with yield and quality of leaves, cocoon and health of the silkworm.

PLOS ONE

Dear Dr. Machado,

Thank you for submitting your manuscript to PLOS ONE. After careful consideration, we feel that it has merit but does not fully meet PLOS ONE’s publication criteria as it currently stands. Therefore, we invite you to submit a revised version of the manuscript that addresses the points raised during the review process.

Registered Report Protocols are still a new kind of report for PLOS One and several of the reviewers were unfamiliar with this kind of manuscript. which led to a couple of incomplete reviews. Reviewers 3 and 4 were well-prepared to examine this particular kind of submission and I encourage the authors to pay close attention to their comments. After going over the reviews myself, it seems clear that the Introduction section could provide more background information for readers to understand the importance of the study, and that it could also be more focused, with less relevant information being removed. Reviewers suggested that there were great ideas expressed in the paper, but the proposed experimental design, may be insufficient to address all of the posed questions. Please respond, point-by-point to the reviewers concerns, paying more attention to the comments made by reviewers 3 and 4.

We look forward to receiving your revised manuscript.

Kind regards,

Adler R. Dillman, Ph.D.

Academic Editor

PLOS ONE

Journal Requirements:

Reviewers' comments:

Reviewer's Responses to Questions

**Comments to the Author**

1. Does the manuscript provide a valid rationale for the proposed study, with clearly identified and justified research questions?

Reviewer #1: No

Reviewer #2: Partly

Reviewer #3: Yes

Reviewer #4: Partly

2. Is the protocol technically sound and planned in a manner that will lead to a meaningful outcome and allow testing the stated hypotheses?

Reviewer #1: Yes

Reviewer #2: No

Reviewer #3: Partly

Reviewer #4: Partly

3. Is the methodology feasible and described in sufficient detail to allow the work to be replicable?

Reviewer #1: No

Reviewer #2: No

Reviewer #3: Yes

Reviewer #4: Yes

4. Have the authors described where all data underlying the findings will be made available when the study is complete?

Reviewer #1: Yes

Reviewer #2: No

Reviewer #3: No

Reviewer #4: No

5. Is the manuscript presented in an intelligible fashion and written in standard English?

Reviewer #1: No

Reviewer #2: No

Reviewer #3: No

Reviewer #4: Yes

6. Review Comments to the Author

You may also provide optional suggestions and comments to authors that they might find helpful in planning their study.

Reviewer #1: The manuscript deals with multiple objectives: a) evaluation of the damages caused by nematodes to the mulberry, b) evaluation of biocontrol agents against nematodes in mulberry, c) evaluation of mulberry leaves from treated plants for feeding silkworms. Some parts of the research could be interesting. However, I cannot understand why silkworms could be damaged when fed with leaves harvested from plants treated with biocontrol agents to the roots 75 days before. Preliminary evidence (or literature) for this should be provided as a rationale for the research conducted here.

It should be noted that the manuscript lacks Results and Discussion. it must be a mistake during the upload.

The abstract must contain the results and the main findings, while the introduction to the problem must be shortened. The following sentences are in contrast one to another: "... Meloidogyne javanica is one of the most important nematode species parasitizing mulberry. To define management strategies, it is important to know if the crop species is damaged by the parasitism of the nematode ...". I would not speak about "disease" caused by a nematode, but damage. The Introduction section contains several interesting information; however, it should be better focused on the topic of the manuscript, and other information should be removed to avoid confusion in the readers. English must be revised by a mother tongue with background in the field of this research, as it is unacceptable in the present form.

Other minor comments are the following.

L77: these sentences are in contrast: "...from the second day after infection, but only in long term experiments,..."

L97: arenaria

L111: list the chemical nematicides registered for application on mulberry in Brazil and, possibly, elsewhere.

L122: the biological control agents Pseudomonas fluorescens and Trichoderma viridae must be introduced. Why did you use them?

L126: you must introduce Artemisia nilagirica extract. Why did you use it?

L186: report the manufacturer

L209: report the formula

L214: Bartlett's test, Shapiro-Wilk's test, and Box-Cox method precede ANOVA, but not the contrary.

L225: abbreviate the genus name after the first occurrence

L230: describe the method for inoculation. Report the dosage per pot.

L364: 5S rRNA is an unusual housekeeping gene: report citation. What is "another constitutive gene"?

Reviewer #2: I have read and reviewed the manuscript “Morus alba: phenotyping for Meloidogyne javanica, biological nematicides assessment and study of these relationships with yield and quality of leaves, cocoon and health of the silkworm” By da Silva et al. The article explains the necessity of being able to grow mulberry in order to cultivate silkworms for commercial purposes. Part of the problem is the susceptibility to parasitic nematodes which interferes with cultivation. Furthermore, the current nematicides also have a negative impact on cultivation.

Major points. The Results and Discussion sections are missing. Consequently, I am recommending a rejection. Furthermore, the writing was poor. The work needs a professional editing, but after the experiments have been done and the completed manuscript written. I have highlighted a few minor points regarding the writing, but every sentence had poor structure.

Minor points:

Introduction:

Sericulture involves the crop of mulberry leaves (Morus alba) [note: please put the genus and species name before leaves].

the silkworm cultivation and the spinning. [note: please put genus and species of the silkworm]

note: always italicize genus and species.

Mulberry is a perennial species with lifespan estimated in about 20 years, easy to

grow and with high rusticity, being cropped in Brazil since 1930. [note: Mulberry is a perennial species with lifespan estimated in about 20 years. Mulberry is easy to

grow and has high rusticity, being cropped in Brazil since 1930.]

In this country, its cultivation stretches between August to May [4] and each tree produces near of four kilograms of leaves per year [7]. [note: In Brazil, mulberry cultivation occurs between August to May [4]. Each tree produces nearly four kilograms of leaves per year [7].

Several Meloidogyne species had been reported parasitizing mulberry worldwide. [note: Several Meloidogyne species have been reported parasitizing mulberry worldwide].

In another study with mulberry, using different population densities of M. Arenaria race 2 [note: M. arenaria].

encasulation? [note: encapsulation?].

Note: I could continue with the grammatical errors.

Materials:

Seedlings will be produced from cuttings taken from mature plants, with buds developed, collected from the germplasm collection maintained by Instituto de Desenvolvimento Rural do Paraná – IDR – IAPAR/ EMATER, located at the municipality of Londrina, Paraná State, Brazil. [note: Seedlings have been produced from cuttings {the work has already been done}.

Seedlings of the three mulberry cultivars, after acclimatization, will be tested

against [note: Seedlings of the three mulberry cultivars, after acclimatization, have been tested

against]

Note: I could continue with the grammatical errors.

Results: There is no Results section

Discussion: There is no discussion section

Reviewer #3: Review comments PONE-D-20-23665

Lines 157-162, the objectives of research seem to be overlapping. The authors need to clarify it.

In the “Phenotyping of mulberry (Morus alba) to Meloidogyne spp” section, the authors need to provide validations why eggs are harvested at “90 days from inoculation” to calculate reproduction factor (RF)? The authors also need to show how the RF is calculated in this section first.

Line 258, is “Ninety DAI” the correct date here?

The experiments in the two sections “Evaluation of biologicalnematicides in the control of Meloidogyne javanica in mulberry cultivar Miura” and “Evaluation of the impacts of the biological nematicides in the yield and quality of mulberry leaves cultivar Miura” can be combined to save time and efforts.

Lines 364-365, the authors need to specify “or another constitutive gene.”

Reviewer #4: The authors want to address the following questions: 1. If mulberry cultivar is damaged by RKN. 2. What is the efficiency of biological nematicides. 3. Whether biological nematicides would improve the quality of leaves or change the nutrient composition, thus change the silkworm production, finally affect the silk industry? In another words, whether biological nematicides have any impact on silk industry.

They designed corresponding experiment to answer those questions.

The first experiment is to phenotyping mulberry cultivar, which I found the word phenotyping a little confusing. They basically want to look at whether three mulberry cultivars are susceptible/resistant to four RKN species. I have some questions about the method here, mostly the reproduce factor (RF) they are going to use. Based on their method, I would say 99% if not 100% of the RF would be greater than 1, which could potentially increase false positives when doing analysis. Also one of the RKN species they will chose is not introduced in the introduction, if in Brazil mostly M. incognita and M. javanica were found, why chose other two that were not reported yet? If they all found in Brazil, then they should be introduced first. They plan to inoculate 1000 eggs per plant, but for their next experiment, they will do 2000 eggs instead, I would suggest keep the inoculation the same.

The second experiment is to evaluate the effect of three biological nematicides on M. Javanica, the method seems standard, I donot have much comment here.

The final series of experiment is supposed to answer the third question (the biggest). They chose two biological nematicides to continue, analyses the nutrition of the leaves, histological analysis of intestinal and glandular tissue of silkworm, and qPCR on couple of the genes that are involved in silk production. The methods itself seem to be standard, but my concern is whether those experiments will generate data to answer the question. For example, I would assume not only two genes involved in silk production. Is it enough to characterize couple genes through qPCR to answer the expression pattern change?

line77 where did you get this? the second day the nematode probably have not sucked nutrient from the plants yet. I think this sentence needs to be edited more carefully, at least cite the paper if it is true.

Line 78 For yield loss, it is very hard to accurately determine.

Line 85-95 those two paragraphs could be condensed and merged

Line 111-114 Any papers to support?

Line 129-134 those two active components are used mostly for insecticides, any citations on biological nematicides effect?

Line 135-143 if there is more background on RKN infecting mulberry cultivars that would be great

line148 Are those four RKN species all reported in affecting mulberry, there is one species that is not introduced anywhere.

lines 150-151 what kind of interaction?

Line158-160 why evaluate 3 and then study two later, what is the criteria for selecting those two?

Line 171 I not sure histological analyzes by itself can support the conclusion on this aspect

line 203 2ml might be too much water, 1ml is a good volume.

Lines 211 any literature for these standards? Since you harvest after 90 days, I suspect 99% of the value will be greater than 1, and that has nothing to do with resistant cultivar, because you have at least 3 to 4 generations. Even on resistant cultivar, the RKN still reproduce, just not as many as susceptible.

Line243 why one-way

Line 264 not very clear which methods you are referring to.

Lines 268-269 please describe more specifically, which are the variables that you will put into multivariate analyzes. Is this typical for nutrient analysis?

Line290 again, the analysis should be more specific

Lines 328 are you using the same sample as previous experiment? How to collect the sample for qPCR. If you use previous sample, my concern is whether the tissue if fresh enough to ensure that no environmental factors interferer with the RNA expression.

7. PLOS authors have the option to publish the peer review history of their article (what does this mean?). If published, this will include your full peer review and any attached files.

Reviewer #1: **Yes: **Giovanni Bubici

Reviewer #2: No

Reviewer #3: No

Reviewer #4: No

---

## [Author Response · Author response to Decision Letter 0]

15 May 2021

Specific responses are as follows:

Reviewer #1: Introduction section was shortened and the sentences indicated as in contrast one to another were excluded in this new short version of this section. In this new version, the topic of the manuscript was focused on the Introduction and all the information that was initially used to better explore the theme was removed; it could be used in the Discussion section in the next manuscript. English was improved throughout the text. 

Other minor comments from reviewer #1 are the following:

L77: these sentences are in contrast: "...from the second day after infection, but only in long term experiments,..." This text was excluded.

L97: arenaria. Excluded 

L111: list the chemical nematicides registered for application on mulberry in Brazil and, possibly, elsewhere. We included in the text the information that there are no chemical nematicides registered for mulberry in Brazil.

L122: the biological control agents Pseudomonas fluorescens and Trichoderma viridae must be introduced. Why did you use them? Excluded. We will not use these organisms. 

L126: you must introduce Artemisia nilagirica extract. Why did you use it? Excluded. We will not use this extract. 

L186: report the manufacturer. ok

L209: report the formula. ok

L214: Bartlett's test, Shapiro-Wilk's test, and Box-Cox method precede ANOVA, but not the contrary. Ok, we changed the order in the text.

L225: abbreviate the genus name after the first occurrence. ok

L230: describe the method for inoculation. Report the dosage per pot. Ok, done.

L364: 5S rRNA is an unusual housekeeping gene: report citation. What is "another constitutive gene"? Analysis done recently by our laboratory showed that the genes initially selected to be used in our study did not amplified adequately. By this reason, we selected other genes from literature that were included in the methodology described in the text. The use of 5S rRNA gene as a housekeeping was a mistake that was corrected in the text; instead, genes RLP3 and alpha-tubulin will be used for this purpose. Table with the primers was corrected and text was modified. 

However, I cannot understand why silkworms could be damaged when fed with leaves harvested from plants treated with biocontrol agents to the roots 75 days before. Preliminary evidence (or literature) for this should be provided as a rationale for the research conducted here. We hypothesize that it could be possible that no deleterious effect will be observed in silkworms and so, mulberry growers could use both biological nematicides in their farms without restrictions. However, it is only an hypothesis and we should prove it to guarantee the safety of the silkworms.

Reviewer # 2. English was improved throughout the text. Minor points indicated regarding structure and grammar were all modified in the text, except when reviewer 2 suggested us to change encasulation by encapsulation, because the word is correct. 

Reviewer # 3. 

Lines 157-162, the objectives of research seem to be overlapping. The authors need to clarify it. We modified the proposition o four work and we think that modifications done in the manuscript could address this question.

In the “Phenotyping of mulberry (Morus alba) to Meloidogyne spp” section, the authors need to provide validations why eggs are harvested at “90 days from inoculation” to calculate reproduction factor (RF)? The authors also need to show how the RF is calculated in this section first. We included the explanation about the calculus of RF values in the Material and Methods section and we altered the evaluation to 120 DAI.

Line 258, is “Ninety DAI” the correct date here? We altered to 120 DAI.

The experiments in the two sections “Evaluation of biological nematicides in the control of Meloidogyne javanica in mulberry cultivar Miura” and “Evaluation of the impacts of the biological nematicides in the yield and quality of mulberry leaves cultivar Miura” can be combined to save time and efforts. Experiments were combined and only one of them, which includes the other, was described to avoid misinterpretations. 

Lines 364-365, the authors need to specify “or another constitutive gene.” The same answer to reviewer 1 could be used here: “Analysis done recently by our laboratory showed that the genes initially selected to be used in our study did not amplified adequately. By this reason, we selected other genes from literature that were included in the methodology described in the text. The use of 5S rRNA gene as a housekeeping was a mistake that was corrected in the text; instead, genes RLP3 and alpha-tubulin will be used for this purpose. Table with the primers was corrected and text was modified.” 

Reviewer # 4. 

The first experiment is to phenotyping mulberry cultivar, which I found the word phenotyping a little confusing. They basically want to look at whether three mulberry cultivars are susceptible/resistant to four RKN species. We decided to evaluate only the main root-knot species for mulberry in Brazil, M. incognita and M. javanica in the phenotyping studies. Although this is a correct word to designate the study that will be done in this section, we agreed with the reviewer and changed to “host reaction of mulberry cultivars to Meloidogyne spp.”

I have some questions about the method here, mostly the reproduce factor (RF) they are going to use. Based on their method, I would say 99% if not 100% of the RF would be greater than 1, which could potentially increase false positives when doing analysis. 

Also one of the RKN species they will chose is not introduced in the introduction, if in Brazil mostly M. incognita and M. javanica were found, why chose other two that were not reported yet? If they all found in Brazil, then they should be introduced first. Both species are found in Brazil parasitizing coffee (M. paranaensis) and other fruit trees (M. enterolobii) and we included these two nematodes aiming to evaluate the reaction of the mulberry cultivars to these two potential pathogens for this crop. But we agreed with the reviewer that the inclusion of these species could lead to misinterpretations of the text and maybe it will not bring us a significant information for our study, as presented. 

They plan to inoculate 1000 eggs per plant, but for their next experiment, they will do 2000 eggs instead, I would suggest keep the inoculation the same. We decided to maintain 1000 eggs per plant in the experiment aiming to evaluate the host reaction of mulberry cultivars to Meloidogyne spp., according to suggested by Silva et al. (2020) for similar studies with coffee plants, also a perennial species. However, in the experiment in which we will evaluate the effect of biological nematicides, we decided to use 5000 eggs per plant. In this case, the higher initial population density could favor the symptoms and damages in the plants and, additionally, nematicides could be submitted to a higher inoculum pressure, commonly found under field conditions. 

The final series of experiment is supposed to answer the third question (the biggest). They chose two biological nematicides to continue, analyses the nutrition of the leaves, histological analysis of intestinal and glandular tissue of silkworm, and qPCR on couple of the genes that are involved in silk production. The methods itself seem to be standard, but my concern is whether those experiments will generate data to answer the question. For example, I would assume not only two genes involved in silk production. Is it enough to characterize couple genes through qPCR to answer the expression pattern change? This concern demonstrated by reviewer 4 is plausible, since there are several genes involved in the silk production reported in literature. Genes that we suggested to use in our study are the more used and they were tested and used in other studies from our lab. Therefore, we initially selected these genes, but we can select other genes during the development of the work, if necessary. 

line77 where did you get this? the second day the nematode probably have not sucked nutrient from the plants yet. I think this sentence needs to be edited more carefully, at least cite the paper if it is true. This sentence was excluded.

Line 78 For yield loss, it is very hard to accurately determine. Sentence was changed.

Line 85-95 those two paragraphs could be condensed and merged. Ok, done.

Line 111-114 Any papers to support? We excluded this sentence from the manuscript.

 Line 129-134 those two active components are used mostly for insecticides, any citations on biological nematicides effect? This sentence was excluded.

Line 135-143 if there is more background on RKN infecting mulberry cultivars that would be great. Unfortunately, information is scarce.

line148 Are those four RKN species all reported in affecting mulberry, there is one species that is not introduced anywhere. We decided to study only M. incognita and M. javanica.

lines 150-151 what kind of interaction? This sentence was excluded.

Line158-160 why evaluate 3 and then study two later, what is the criteria for selecting those two? We decided to work with only two biological nematicides, with different modes of action, one bacteria that protects the roots against the nematode penetration as a main effect and one fungi that has a direct effect capturing juveniles and parasitizing eggs.

Line 171 I not sure histological analyzes by itself can support the conclusion on this aspect. We think that probably no effect will be observed and that the gland patterns observed in the check will also be observed in the treatments. But, considering the volume of samples and cuts that will be done, if any variation could occur, we will be able to verify it. 

 line 203 2ml might be too much water, 1ml is a good volume. Considering that 2 ml correspond to only 0.23% from the volume of a 900 ml-capacity pot, we consider that this is not too much water. Therefore, except by a personal opinion from reviewer, we cannot find any justificative to reduce the volume of the inoculum. Even because the spatial distribution of the inoculum in the pot is important to guarantee the uniform infestation. 

Lines 211 any literature for these standards? Since you harvest after 90 days, I suspect 99% of the value will be greater than 1, and that has nothing to do with resistant cultivar, because you have at least 3 to 4 generations. Even on resistant cultivar, the RKN still reproduce, just not as many as susceptible. Mulberry is a perennial crop and earlier evaluations of the host reaction to nematodes could lead to erroneous results. If we consider an annual plant species, as Phaseolus vulgaris, whose reproductive cycle is around 65 days, it is plausible to consider an evaluation at 30 days after inoculation, as proposed by Oostenbrink (1966), since at 30 DAI, we have 40% of the total cycle of this plant. Mulberry plants take up one year from cutting planting to production, considering that mulberry flowering occur at spring and fructifying at summer. So, we can consider that the reproductive cycle of mulberry is completed around one year and, therefore, the evaluation done 3 to 4 months after the inoculation seems to be more plausible for us. Our group have been working with coffee plants and our experience with this crop shows that resistant cultivars maintain this characteristic also in evaluation done at 150 DAI. 

Line243 why one-way. The variables are dependent.

Line 264 not very clear which methods you are referring to. Nematodes will be extracted from roots according to Boneti and Ferraz (1981) methodology, as described for the host reaction experiment, from plants whose shoot parts were used as feed source to silkworms. 

Lines 268-269 please describe more specifically, which are the variables that you will put into multivariate analyzes. Is this typical for nutrient analysis? We described in detail in the text all variables that will be analyzed. The multivariate analysis allows us a global view of the effect of treatments based in the set of variables.

Line290 again, the analysis should be more specific. The analysis was better described in the text. 

Lines 328 are you using the same sample as previous experiment? How to collect the sample for qPCR. If you use previous sample, my concern is whether the tissue if fresh enough to ensure that no environmental factors interferer with the RNA expression.

A detailed methodology explaining how the tissues will be extracted and stored was added in this section to clarify this question.

---

## [Editor Report · Decision Letter 1]

27 May 2021

Morus alba: host reaction for Meloidogyne javanica, biological nematicides assessment and study of these relationships with yield and quality of leaves, cocoon and health of the silkworm.

PONE-D-20-23665R1

Dear Dr. Machado,

We’re pleased to inform you that your manuscript has been judged scientifically suitable for publication and will be formally accepted for publication once it meets all outstanding technical requirements.

Kind regards,

Adler R. Dillman, Ph.D.

Academic Editor

PLOS ONE

Additional Editor Comments (optional):

Thank you for addressing the reviewers concerns in the revision. The revision is significantly improved.

There are a few minor textual suggestions I would make that I think would improve clarity.

For example, "encasulation," which you have used on line 194 to mean "reach the pupal stage," is not a word, to my knowledge. I only find reference to it in other papers as a typo where the word "encapsulation," is meant. However, in the context used on line 194 "encapsulation" does not mean pupation.

Line 27: change "damages" to damage.
---

## [Editor Report · Acceptance letter]

31 May 2021

PONE-D-20-23665R1 

*Morus alba*: host reaction for *Meloidogyne javanica*, biological nematicides assessment and study of these relationships with yield and quality of leaves, cocoon and health of the silkworm. 

Dear Dr. Machado:

I'm pleased to inform you that your manuscript has been deemed suitable for publication in PLOS ONE. Congratulations! Your manuscript is now with our production department. 

Kind regards, 

on behalf of

Dr. Adler R. Dillman 

Academic Editor

PLOS ONE